# COVID-19 Breakthrough Infections and Transmission Risk: Real-World Data Analyses from Germany’s Largest Public Health Department (Cologne)

**DOI:** 10.3390/vaccines9111267

**Published:** 2021-11-02

**Authors:** Lea Hsu, Barbara Grüne, Michael Buess, Christine Joisten, Jan Klobucnik, Johannes Nießen, David Patten, Anna Wolff, Gerhard A. Wiesmüller, Annelene Kossow, Julia Hurraß

**Affiliations:** 1Public Health Department Cologne, Infektions- und Umwelthygiene, 50667 Köln, Germany; barbara.gruene@stadt-koeln.de (B.G.); michael.buess@stadt-koeln.de (M.B.); c.joisten@dshs-koeln.de (C.J.); jan.klobucnik@dozent.ism.de (J.K.); johannes.niessen@stadt-koeln.de (J.N.); wilfried.patten@stadt-koeln.de (D.P.); anna.wolff@stadt-koeln.de (A.W.); gerharda.wiesmueller@stadt-koeln.de (G.A.W.); annelene.kossow@stadt-koeln.de (A.K.); julia.hurrass@stadt-koeln.de (J.H.); 2Institute for Occupational Medicine and Social Medicine, University Hospital, Medical Faculty, RWTH Aachen University, 52056 Aachen, Germany; 3Department for Physical Activity in Public Health, Institute of Movement and Neurosciences, Am Sportpark Müngersdorf 6, German Sport University Cologne, 50933 Cologne, Germany; 4Institute of Hygiene, University Hospital Muenster, 48149 Münster, Germany

**Keywords:** COVID-19, vaccination, breakthrough infections, SARS-CoV-2 pandemic, SARS-CoV-2 transmission

## Abstract

Background and Methods: Vaccination is currently considered the most successful strategy for combating the SARS-CoV-2 virus. According to short-term clinical trials, protection against infection is estimated to reach up to 95% after complete vaccination (≥14 days after receipt of all recommended COVID-19 vaccine doses). Nevertheless, infections despite vaccination, so-called breakthrough infections, are documented. Even though they are more likely to have a milder or even asymptomatic course, the assessment of further transmission is highly relevant for successful containment. Therefore, we calculated the real-world transmission risk from fully vaccinated patients (vaccination group, VG) to their close contacts (CP) compared with the risk from unvaccinated reference persons matched according to age, sex, and virus type (control group = CG) utilizing data from Cologne’s health department. Results: A total of 357 breakthrough infections occurred among Cologne residents between 27 December 2020 (the date of the first vaccination in Cologne) and 6 August 2021. Of the 979 CPs in VG, 99 (10.1%) became infected. In CG, 303 of 802 CPs (37.8%) became infected. Factors promoting transmission included non-vaccinated status (*β* = 0.237; *p* < 0.001), male sex (*β* = 0.079; *p* = 0.049), the presence of symptoms (*β* = −0.125; *p* = 0.005), and lower cycle threshold value (*β* = −0.125; *p* = 0.032). This model explained 14.0% of the variance (corr. *R*^2^). Conclusion: The number of transmissions from unvaccinated controls was three times higher than from fully vaccinated patients. These real-world data underscore the importance of vaccination in enabling the relaxation of stringent and restrictive general pandemic control measures.

## 1. Introduction

Severe acute respiratory syndrome (SARS-CoV-2) vaccination is the most critical measure for containing the progression of the COVID-19 pandemic worldwide. In Germany, vaccination with the Pfizer-BioNTech (Mainz, Germany) BNT162b2 mRNA vaccine began on 27 December 2020 [1]. Vaccination with the Oxford-AstraZeneca (Cambridge, UK) ChAdOx1 (AZD1222), Moderna mRNA-1273 (Cambridge, MA, USA) and Johnson & Johnson (New Brunswick, NJ, USA) INJ-7843735 (Ad26.COV2) vaccines followed shortly thereafter. In September 2021, approximately 63% of Germans over the age of 18 years were fully vaccinated [2]. After complete vaccination (≥14 days after receipt of all recommended COVID-19 vaccine doses), protection against infection reportedly reached up to 95% in short-term clinical trials [3]. In addition to significantly decreasing hospitalizations and deaths, real-world data also indicate that vaccination reduces the chances of contracting COVID-19 and increases the likelihood that the infections that do occur will be asymptomatic or mild [3,4,5]. Furthermore, significantly milder courses and reduced mortality [6] have been described in the context of so-called breakthrough infections (i.e., infections that occur despite vaccination). Levine-Tiefenbrun et al. [7] postulated that beyond substantial protection against infection, vaccination might reduce the viral load in breakthrough infections and thereby suppress further transmission. They found that the viral load was substantially reduced for infections occurring 12–37 days after the first BNT162b2 dose. These initially positive observations were contradicted by findings indicating that viral variants could reduce the protective effects of vaccination. In particular, the Delta variant (B.1.617.2) has shown comparable viral loads in unvaccinated individuals versus vaccinated individuals with breakthrough infections [8]. However, Chia et al. [9] described a faster decline in viral load in vaccinated people infected with the Delta variant.

Knowledge of transmission risk is essential in order to relax general lockdown measures, which are sometimes considerably restrictive and psychologically stressful, and to gradually return to pre-pandemic life. To evaluate the infectiousness of an individual, cycle threshold (Ct) values are used as an indirect marker of the viral load. Other possibilities for measuring transmission risk include cohort or case-control studies. To interrupt chains of infection in Germany, infected persons and their close contacts (contact persons, CP) are recorded according to the legal regulations for the control of infectious diseases under the Infection Protection Act. Initial analyses indicated that transmission risk for vaccinated IPs was less than half that of unvaccinated IPs in terms of breakthrough infections with the delta variant [10]. However, onward transmission of other variants remained unclear when taking Ct values and vaccination status into account. Therefore, we examined data from the largest German public health department in Cologne regarding transmission from fully vaccinated patients to fully vaccinated CPs compared with an unvaccinated control group (CG).

## 2. Methods

### 2.1. Study Design

The Cologne public health department is the largest public health department in Germany and serves a population of 1,083,498. It registered all infected persons detected via PCR (polymerase chain reaction) test and their relevant contacts in the city since the first occurrence of SARS-CoV 2 in Cologne in March 2020. A contact of a COVID-19 IP is any person who has had close exposure to a confirmed COVID-19 case (<1.5 m) for more than 10 min without a mask within two days of symptom onset in the index case to 14 days after symptom onset [11].

To reliably record COVID-19 cases, the affected individuals are contacted by telephone and placed in quarantine by a standardized procedure (Figure 1). As part of this initial contact, data on age, sex, risk factors for a severe outcome of the disease (e.g., obesity, diabetes, cardiovascular diseases, pulmonary diseases, chronic liver diseases, cancer, immunodeficiency), symptom onset, vaccinations (type and date), route of transmission, and contact persons [12] during the infectious period are recorded in the specially developed software program called DiKoMa [13]. CPs are also registered, and their symptoms are recorded in this database. PCR testing is provided for CPs when symptoms occur. After the Delta variant became common in Cologne (first occurrence in April 2021; Figure 2), mandatory free testing at the end of the quarantine period was also introduced for CPs and IPs.

### 2.2. Study Population

This study included people living in Cologne who had tested positive for SARS-CoV-2 via PCR and who had been fully vaccinated between 27 December 2020 and 6 August 2021 (Figure 3) and their respective CPs also residing in Cologne (*n* = 357). We excluded all IPs and CPs who did not live in Cologne, and those whose CPs did not live in Cologne were excluded.

IPs from the same observation period who had not yet received a full or partial vaccination (*n* = 30,465) served as the CG. Each patient in the vaccination group (VG) was randomly assigned to a patient from the CG matched by age, sex, and virus type (variant of concern (VoC) or wild type).

### 2.3. Matching

We selected non-vaccinated controls from the DiKoMa registry of patients with PCR-confirmed COVID-19 from the same observation period (*n_total_* = 30,465; *n_unvaccinated_* = 27,457). Each patient in the vaccination group (VG) was randomly matched 1:1 to a SARS-CoV-2 positive non-vaccinated patient (CG). Age, sex, and virus type (a variant of concern (VoC) or wild type) were chosen as matching criteria since they may influence the immune response to vaccination and transmissibility [14,15,16]. IPs who were not fully vaccinated (*n_incomplete vaccinated_* = 2035) were excluded from this analysis.

### 2.4. Laboratory Analyses

Accredited laboratories performed real time (RT) PCR tests. For every sample, the Ct value at the time of the smear was also reported. A Ct value under the threshold of 30 was used to determine transmissibility [17]. Because whole genome sequencing is too costly considering the high number of samples, the Cologne laboratories detected known mutations using RT-PCR assays that target these key mutations. For samples that could not be evaluated, the currently prevailing VoC was assumed to be present (up to and including week 8 wild type, week 9–24 Alpha, and week 25–31 Delta) [18].

### 2.5. Transmission Risk

Disease transmission was assumed if a CP of an IP also tested positive for SARS-CoV-2 within the 10 to 14-day post-contact incubation period. During the incubation period, the Cologne public health department also interviewed the CPs regarding their risk profiles and COVID-19-specific symptoms and maintained a digital symptom diary. If typical symptoms occurred, the health department organized testing; they refrained from systematic testing of asymptomatic people. Every infection that could be traced to a previous case (source case) was also recorded in the IP’s digital case file. Thus, initially reported CPs were recorded, and infections that could clearly be traced to an IP were tracked.

### 2.6. Statistical Analyses

Regarding descriptive statistics, absolute and relative frequencies were calculated for categorical variables, and means and standard deviations (SDs) were calculated for continuous variables. Associations between participant characteristics (e.g., age and sex) of the two groups and outcomes were examined using *χ^2^* tests or independent *t*-tests.

Linear regression was used to examine the influence of vaccination (yes = 1, no = 2), age (in years), sex (female = 1, male = 2), virus type (variants others than B.1.617.2 = 1 vs. B.1.617.2 = 2), presence of risk factors for severe disease (no = 1, yes = 2), presence of symptoms (yes = 1, no = 2), and Ct value on the number of infected CPs relative to the total number of CPs per IP.

Infections in CPs were analyzed using binary logistic regression (CP infected? yes/no). In addition to age and sex, the vaccination status of the CP and the source case were considered. Odds ratios (ORs) and 95% confidence intervals (CIs) were calculated in each case, and the model fits were checked using pseudo-*R*^2^ (Nagelkerke’s *R*^2^) values.

A *p*-value below 0.05 was considered significant. Analyses were performed using SPSS version 27.0 (IBM, Armonk, NY, USA).

## 3. Results

### VG versus CG

In total, 357 fully vaccinated IPs (VG) had 979 Cologne CPs and were matches with 357 unvaccinated IPs (CG) and their respective 802 Cologne CPs (Figure 3). No differences in age, sex, or presence of a variant or wild type were evident (Table 1). Individuals in the VG were significantly less symptomatic than individuals in the CG (*p* < 0.001; Table 1). In both groups, Ct values of symptomatic IPs were significantly lower than those of asymptomatic IPs (both *p* < 0.001; data not shown).

Of the VG, 80.1% had been vaccinated with Pfizer–BioNTech *(n* = 286), 8.4% with Johnson & Johnson (*n* = 30), 3.9% with Oxford–AstraZeneca (*n* = 14), 3.1% with Moderna (*n* = 11), 0.6% with Sputnik or Sinopharm (*n* = 2) and 3.9% with a combination vaccine (*n* = 14). On average, the vaccination interval was 62.4 ± 35.2 days (a range of 14 to 188 days).

Of the 979 close CPs in the VG, 99 (10.1%) became infected; in the matched CG, 303 of 802 CPs (37.8%) became infected. On average, 2.74 ± 3.11 CPs (range = 1–30) were reported in the VG and 2.24 ± 1.63 (range = 1–11; *p* = 0.008; Table 2) in the CG. In the VG, 0.27 ± 0.69 CPs were infected per IP (range = 0–5) in comparison with 0.79 ± 1.10 CPs per IP in the CG (range = 0–6; *p* < 0.001; Table 2). Regarding the number of CPs reported, 0.13 ± 0.31 CPs were infected and 0.37 ± 0.43 CPs were infected in the CG (*p* < 0.001; Table 2). The Ct value was significantly higher in the VG at 29.5 ± 7.5 than in the CG at 25.7 ± 6.6 (*p* < 0.001; Table 2) and lower in patients infected with the Delta variant (Table 2).

Table 3 shows the final model of factors influencing the number of infected CPs relative to the total number of CPs per IP. Unvaccinated individuals (*β* = 0.237; *p* < 0.001) and men (*β* = 0.079; *p* = 0.049) exhibited a higher ratio of infected CPs to total CPs. The presence of symptoms (*β* = −0.125; *p* = 0.005) and lower Ct score (*β* = −0.125; *p* = 0.032) were also risk factors for a higher proportion of infected CPs in relation to total CPs per IP. The model explained 14.0% (corr. *R*^2^) of the variance (Table 3).

Vaccinated versus unvaccinated contacts: impact on transmission.

A total of 439 CPs were fully vaccinated, and 1182 CPs were unvaccinated. Data from partially vaccinated individuals were not included in this analysis. Regardless of the transmission route, infected CPs were significantly less likely to be fully vaccinated (*p* < 0.001; Table 4).

The Ct value of unvaccinated infected CPs (*n* = 270) was not significantly different (25.6 ± 6.5) from that of vaccinated infected CPs (26.2 ± 7.3; *n* = 56; *p* = 0.599).

However, binary logistic regression revealed that the rate at which CPs were infected per IP was 79% lower when the source case was fully vaccinated than when the source case was unvaccinated (*p* < 0.001). Age, sex, and vaccination status of the CP had no influence (Appendix A, Figure 4).

## 4. Discussion

Complete vaccination reduced the number of infections in CPs by more than two-thirds in our analysis. Non-vaccination, male gender, the presence of symptoms, and lower Ct scores were risk factors for a higher proportion of infected CPs in relation to total CPs. Previous analysis of England’s Household Transmission Evaluation Dataset (HOSTED) has shown a reduction in transmission within the households of infected patients who had been vaccinated 21 or more days before testing positive by approximately 40% to 50% compared with the households of unvaccinated infected patients [19]. More than 90% of their population had been only partially vaccinated by that time. A study from Belgium, based on a similar contact tracing system, also described significant protection against infection after a close contact; however, the study only examined fully vaccinated IPs. Moreover, additional protection against transmission was demonstrated when the CP was also fully vaccinated [20]. A retrospective analysis from England confirmed lower transmission to close contacts among vaccinated persons for the Alpha variant and the same effect but somewhat less pronounced for the Delta variant [21]. The Ct values played a lesser role in this study. Our study further showed that Ct values, as a marker of infectivity, were significantly higher in the group of vaccinated than in the group of unvaccinated IPs and lower with the occurrence of the Delta variant in both vaccinated and unvaccinated IPs. Additionally, the Ct value was negatively correlated with the number of infected CPs per IP. Most previous studies have described comparable Ct values in vaccinated and unvaccinated IPs [8,22,23]. However, it must be noted that our study analyzed Ct levels only once, during PCR testing. Moreover, such a correlation did not appear in the group of infected CPs. These CPs were scheduled to be tested because they had traceable contact with an IP, but most source cases included in our analysis could not report a traceable contact (>60% in both groups; data not shown). To accurately determine the reliability of the Ct value, serial studies would need to be performed; however, these were not possible in this setting. Furthermore, analyses of Ct values may lead to flawed conclusions, as it is presently unknown whether (and to what extent) variation in Ct reflects variation in viral load or gene expression [24].

### Strength and Limitations

The greatest strength of this analysis is the Cologne public health department’s systematic and almost complete data collection, which allows for a reliable statement on real-world transmission risk. One limiting factor is that the number of IPs infected with newer variants, especially the number of fully vaccinated IPs infected with the Delta variant, was still small. In addition, a statement regarding the protection offered by different vaccines is not yet possible because mainly Pfizer-BioNTech was used during the timeframe examined by the study. Further factors not yet considered are the influence of age on breakthrough infections and the possibility that vaccine effectiveness may decrease six months after vaccination.

As the focus of this study was the risk of transmission, we considered the occurrence of symptoms but not the clinical course. In general, vaccinated people not only have protection against infection, but also a less severe course of the illness. Rymksi et al. [25] confirmed these facts in their analysis, but pointed out that certain risk groups, especially immunosuppressed ones, may become severely ill due to a lower immune response or to being vaccine non-responders. The course of the antibodies was also not taken into account in our study since systematic screening for antibodies in the general population does not exist in Germany. Favresse et al. [26] described a decrease in antibodies after three months. Thomas et al. [27] showed a decrease in the vaccine efficacy of 6% every two months. In our study, we analyzed the influence of the vaccination distance on the number of infected CPs in the VG via linear regression (data not shown). In the final model, the vaccination distance showed a significant but weak correlation (*β* = 0.136; *p* = 0.015). Presumably, our observation period is still too short to obtain valid results here. Therefore, further clinical investigations are necessary considering the available vaccines, timespan, or VoCs, as well as close monitoring of the antibodies to assess the time for booster vaccinations and thus reduce the risk of supposed breakthrough infections.

## 5. Conclusions

Fully vaccinated IPs showed a lower viral load, less frequent symptoms, and, above all, a significantly reduced risk of transmission. To effectively end the pandemic, vaccination should be encouraged until herd immunity may be achieved. Until then, to avoid breakthrough infections and the emergence of further VoCs, general protective measures such as contact restrictions and mask wearing should remain in place, especially for the unvaccinated population.

## Figures and Tables

**Figure 1 vaccines-09-01267-f001:**
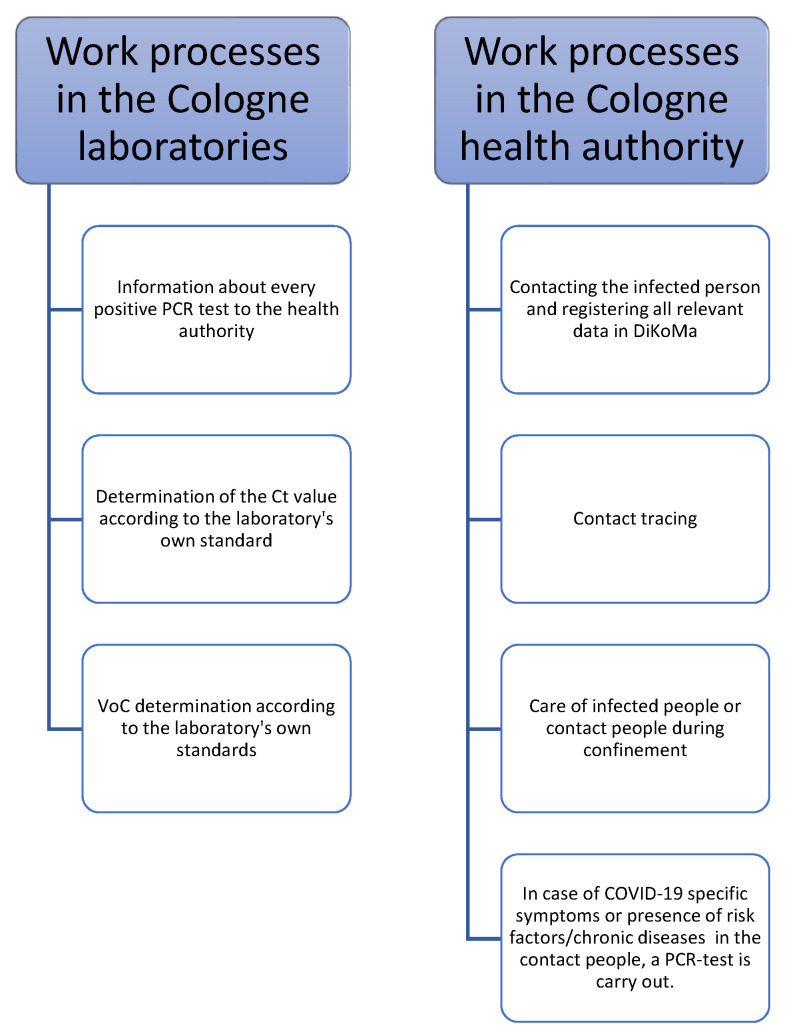
Operational procedures in the Cologne Health Department and Cologne laboratories.

**Figure 2 vaccines-09-01267-f002:**
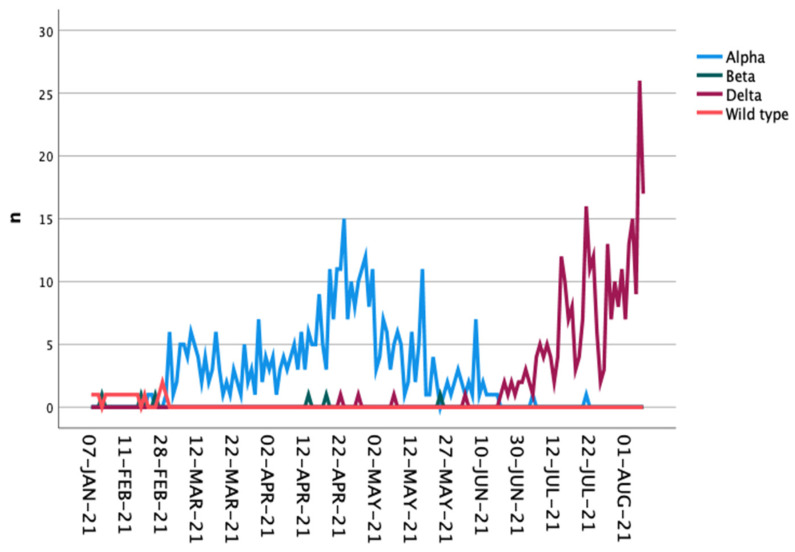
Occurrence of the different virus variants during the observation period in Cologne.

**Figure 3 vaccines-09-01267-f003:**
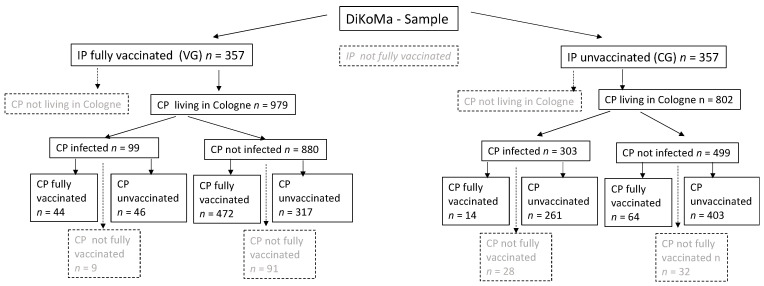
Flow chart of the study population; VG = vaccination group, CG = comparison group, CP = contact person; grey = data not shown.

**Figure 4 vaccines-09-01267-f004:**
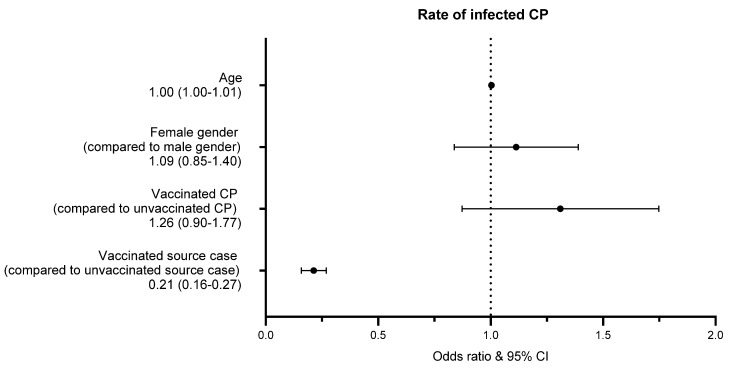
Factors influencing transmission of CPs; binary logistic regression.

**Table 1 vaccines-09-01267-t001:** Demographic characteristics, and differences between VG (vaccinated group) and CG (control group). * *χ^2^*-test; ** unpaired *t*-test.

	VG *n* (%)	CG *n* (%)	*p*-Value
Female	231 (64.7%)	231 (64.7%)	n.s. *
Male	126 (35.3%)	126 (35.3%)	
Age (years) Mean ± SD	48.6 ± 22.1	46.7 ± 21.0	n.s. **
VoC/virus type			
Wild type	9 (2.5%)	9 (2.5%)	n.s. *
Alpha	202 (56.6%)	202 (56.6%)	
Beta	3 (0.8%)	3 (0.8%)	
Delta	143 (40.1%)	143 (40.1%)	
Symptoms			
Yes	147 (41.2%)	287 (77.9%)	<0.001 *
No	210 (58.8%)	79 (22.1%)	

**Table 2 vaccines-09-01267-t002:** a: Ct value (cycle threshold value), total CPs (contact persons) per IP (index person), infected CPs total and in relation to total CPs per IP in VG versus CG. * unpaired *t*-test. b: Differences between the Ct-value (cycle threshold value) in the VG (vaccinated group) and CG (control group) regarding the Alpha and Delta variants. Wild type and Beta were excluded because of their small numbers. * unpaired *t*-test.

a					
	Group (*n*)		Mean	SD	*p*-Value *
Ct value	VG (300)		29.5	7.5	<0.001
	CG (287)		25.7	6.6	
Number of CPs per IP	VG (357)		2.74	3.11	0.008
	CG (357)		2.24	1.63	
Number of infected CPs per IP	VG (357)		0.27	0.69	<0.001
	CG (357)		0.79	1.06	
Number of infected CPs to total number of CPs per IP	VG (357)		0.13	0.31	<0.001
	CG (357)		0.37	0.43	
**b**					
**Ct-value**	**VoC**	** *n* **	**Mean**	**SD**	***p*-Value ***
VG	Alpha	156	33.1	6.0	<0.001
	Delta	133	25.0	6.7	
CG	Alpha	154	26.9	6.4	<0.001
	Delta	124	24.1	6.4	

**Table 3 vaccines-09-01267-t003:** Factors influencing the number of infected CPs relative to total CPs.

Model		Non-Standardized Coefficients		Standardized Coefficients	Sig.	95% Confidence Interval	
		Regression Coefficient (B)	Std. Error	*β*		Lower Limit	Upper Limit
Baseline model	Age (years)	0.002	0.001	0.087	0.034	0.00	0.00
	Gender (female = 1; male = 2)	0.065	0.033	0.080	0.046	0.00	0.13
	Ct value	−0.006	0.002	−0.115	0.011	−0.01	0.00
	B.1.617.2 versus others	0.030	0.036	0.038	0.396	−0.04	0.10
	VG (1) versus CG (2)	0.191	0.033	0.243	<0.001	0.13	0.26
	Symptomatic (yes = 1; no = 2)	−0.093	0.037	−0.116	0.013	−0.17	−0.02
Final model	Age	0.001	0.001	0.079	0.049	0.00	0.00
	Gender (female = 1; male = 2)	0.070	0.032	0.085	0.032	0.01	0.13
	Ct value	−0.007	0.002	−0.125	0.005	−0.01	0.00
	VG (1) versus CG (2)	0.186	0.033	0.237	<0.001	0.12	0.25
	Symptomatic (yes = 1; no = 2)	−0.100	0.036	−0.125	0.006	−0.17	−0.03

**Table 4 vaccines-09-01267-t004:** Influence of CP (contact person) vaccination status on infection status; * Chi-squared-test.

	Vaccination Status		*p*-Value
	*Unvaccinated*	*Fully Vaccinated*	
Non-infected CP	875 (74.0%)	381 (86.8%)	<0.001 *
Infected CP	307 (26.0%)	58 (13.2%)	

## Data Availability

The data presented in this study are available on request from the corresponding author.

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
