# Peer review of "COVID-19 Breakthrough Infections and Transmission Risk: Real-World Data Analyses from Germany’s Largest Public Health Department (Cologne)"

_vaccines, 2021, doi:10.3390/vaccines9111267_

Round 1

Reviewer 1 Report

In this paper the Authors aim to evaluate COVID-19 Breakthrough Infections and Transmission Risk: Real-World Data Analyses from Germany’s Largest Public Health Department. A comprehensive and extensive literature review of the NCBI database PubMed was also carried out. The article was well conducted and it is interesting in its fields. It is a well-structured paper, written in good English and the References are up dated. 

Minor issues:

The during COVID-19 pandemic, the clinical evaluation of patients is deeply changed. The use of telemedicine and remote counselling, in fact, has gained great importance during covid 19 pandemic also in surgical fileds. In the “discussion” section I suggest to better analyze this topic.

Author Response

In this paper the Authors aim to evaluate COVID-19 Breakthrough Infections and Transmission Risk: Real-World Data Analyses from Germany’s Largest Public Health Department. A comprehensive and extensive literature review of the NCBI database PubMed was also carried out. The article was well conducted and it is interesting in its fields. It is a well-structured paper, written in good English and the References are up dated. 

Minor issues:

The during COVID-19 pandemic, the clinical evaluation of patients is deeply changed. The use of telemedicine and remote counselling, in fact, has gained great importance during covid 19 pandemic also in surgical fields. In the “discussion” section I suggest to better analyze this topic.

Thank you for this important reference. However, the focus of this paper is mainly on transmission, not on clinical care. We therefore ask for your understanding that we have not currently included the aspect of telemedical care in the discussion.

Reviewer 2 Report

The manuscript described a study analyzing the transmission risk from vaccinated and unvaccinated individuals. The study was performed in Cologne, Germany. The findings are encouraging. The main one is showing that this risk is three times lower in the case of the vaccinated group. The manuscript is written in good English. Some revisions are, however, required to strengthen the Introduction and Discussion.

  1. L116-17: I suggest reformulating this sentence. 95% efficacy against infection was noted only for approved mRNA vaccines and only in short-term clinical trial observations. I would emphasize that this ‘up to 95%’ value originates from short-term clinical trials observations, does not include the decline in antibodies occurring within a few months after vaccination and the emergence of more transmissible variants.
  2. L20: just write: „(IP)”
  3. L21: “vaccination group, VG)”
  4. L24: “(control group, CG)”
  5. Extend keywords by adding: breakthrough infections, SARS-CoV-2, pandemic
  6. L46: asymptomatic or mild
  7. L51: Elaborate more on this. Some observations indicated similar Ct values in unvaccinated and vaccinated, both infected with delta. This has been a clear change in what was reported earlier – higher CT values in matched vaccinated individuals.

https://www.nature.com/articles/s41591-021-01316-7

  1. However, one should note that Ct values do not necessarily inform on viral load, and by that – transmission - as they may be due to both genomic or subgenomic RNA. This is well explained here:

https://www.cell.com/trends/microbiology/fulltext/S0966-842X(21)00208-0

  1. Finally, the analysis of the dynamic of Ct values in delta-infected vaccinated and unvaccinated individuals shows that they have similar values at the beginning, but the values go high after five days in the former group.

https://www.medrxiv.org/content/10.1101/2021.07.28.21261295v1

  1. I believe having the above points, and references in the Introduction will strengthen the rationale for your study.
  2. I believe providing some background on a decline in serum antibody levels a few months after completing the vaccination regime would be beneficial. This decline likely translates into an increased risk of breakthrough infection (but not necessarily breakthrough disease).

https://www.ncbi.nlm.nih.gov/pmc/articles/PMC8300930/

https://www.medrxiv.org/content/10.1101/2021.07.28.21261159v1

  1. An additional issue is an infection in non-responder groups. Some groups, especially with immune deficiency, may not respond at all or respond significantly more weakly to the vaccination, making them vulnerable to infection (often without knowing if not tested for IgG antibodies):

https://jasn.asnjournals.org/content/32/9/2153.long

https://www.mdpi.com/2076-393X/9/7/781

  1. L80: Provide more data on B.1.617.2 prevalence in the studied region during the observed period. Particularly, provide a period of dominance of alpha and then delta variant.
  2. I would appreciate showing the Ct values for delta vs. alpha variant. You can report in the text and provide a statistical comparison and then discuss it.
  3. L149-155: This part is more suitable for methodology description, and here just report the findings of linear regression.
  4. My concern is the wide timeline during which the vaccinated individuals were vaccinated, between 217 Dec 20 and 6 Aug 2021. As we know, the serum levels of IgG antibodies wane over time from the last vaccination. Israeli data shows that efficacy against infection is much lower for those vaccinated at the beginning compared to those vaccinated later. https://www.gov.il/BlobFolder/reports/vaccine-efficacy-safety-follow-up-committee/he/files_publications_corona_two-dose-vaccination-data.pdf

How could it affect your results?

  1. I believe that discussing the recent observations on transmission of alpha and delta reported from the UK should be included in the Discussion: https://www.medrxiv.org/content/10.1101/2021.09.28.21264260v1
  2. L207: Not sure if herd immunity is even possible. https://www.nature.com/articles/d41586-021-00728-2 Making COVID-19 endemic seem likely.

Author Response

(The authors gave the same response as above.)

Reviewer 3 Report

the abstract is very confusing. English revision is needed. Please make clear the aims, the methods, and principal results. 

it is not clear for which variables cases and controls were matched. please clarify.

At the beginning of the discussion, the authors should summarize the main results. It is now missing. Morevoer, comparison with previous similar studies should be performed in order to determine external validity of the study.

Strenghts and limitation are not discussed. Please add

conclusions should highlight the pubic health impact of these data.

Author Response

The abstract is very confusing. English revision is needed. Please make clear the aims, the methods, and principal results. 

We would first like to apologize for the confusion. The manuscript was intensively revised and then reviewed again by a native speaker. Following your advice, the abstract was restructured and the aims were presented more clearly. 

It is not clear for which variables cases and controls were matched. please clarify.

Thank you very much for this comment. We have now described the matching process more clearly and substantiated the criteria for selection with the relevant literature. 

At the beginning of the discussion, the authors should summarize the main results. It is now missing.

That is a valuable aspect. We have now presented the main results at the beginning of the discussion.

Morevoer, comparison with previous similar studies should be performed in order to determine external validity of the study.

Many thanks for this hint too; we have sifted through many current studies and added  them when relevant, especially in the introduction and discussion.

Strenghts and limitation are not discussed. Please add

The strengths and limitations have been added. We have added a corresponding section.

Conclusions should highlight the public health impact of these data.

Thank you very much, we have added the public health impact more clearly in the whole discussion. 

Round 2

Reviewer 1 Report

In my opinion is suitable for publication in its present form

Reviewer 2 Report

The authors did not respond to my comments. Instead, they have pasted the response to another reviewer. Please provide a correct response.

Round 3

Reviewer 2 Report

Dear Authors

Thank you for your efforts in revising the manuscript. I appreciate the new figure showing the distribution of variants. I think this is a very fine study and manuscript, providing valuable data of post-authorization real-world monitoring of COVID-19 vaccines. I don't have any further comments.